# Joint Semantic and Strategy Matching for Persuasive Dialogue

**Chuhao Jin[1], Yutao Zhu[1], Lingzhen Kong[1], Shijie Li[2], Xiao Zhang[1], Ruihua Song[1*],**
**Xu Chen[1], Huan Chen[2], Yuchong Sun[1], Yu Chen[2], Jun Xu[1]**

[1]Gaoling School of Artificial Intelligence, Renmin University of China, Beijing, China
[2]Meituan, Beijing, China
{jinchuhao,ytzhu,konglingzhen,rsong}@ruc.edu.cn,
{lishijie11,chenhuan15,chenyu17}@meituan.com

## Abstract

Persuasive dialogue aims to persuade users to achieve specific goals through conversations. While previous models have achieved notable successes, they mostly rely on matching utterance semantics and neglect an important aspect: the strategies of a conversation, such as *emotional-appeal* and *foot-in-door*. In contrast to utterance semantics, conversation strategies are high-level concepts, which can be informative and provide complementary information, contributing to more effective persuasion. This paper proposes a novel persuasion framework that combines the modeling of conversation semantics and strategies. To accomplish this objective, we design a BERT-like module and an auto-regressive predictor that match the semantics and strategies, respectively. Experimental results indicate that our proposed approach can significantly improve the state-of-the-art baseline by 5% on a small dataset and 37% on a large real-world dataset in terms of Recall@1. The online evaluation shows that our approach improves the ultimate goal of persuasion in real-world applications.

## 1 Introduction

Persuasive dialogue has recently attracted increasing attention from the research community and shown great potential in real-world scenarios. It aims to simulate the persuaders to communicate with the users for achieving a given goal (*e.g.*, raising charitable donations). As demonstrated in Figure 1, the agent (left side) must not only introduce the charitable organization, but also consider the user's emotions and encourage a reasonable donation amount in order to persuade the user (right side) to donate. Compared with the other conversation tasks, persuasive dialogue can be particularly challenging as it involves not only keeping the user

---

This work was performed when Chuhao Jin and Lingzhen Kong were visiting Meituan as research interns.
*Corresponding author.

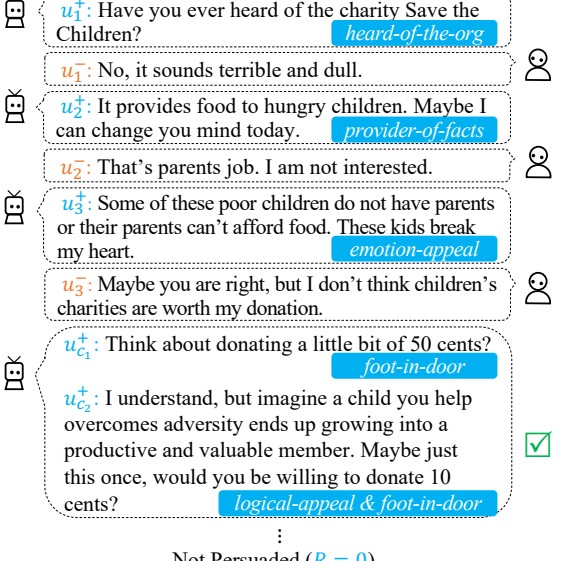

Figure 1: Part of a persuasive dialogue session from *PersuasionForGood*, with strategies labelled in color.

engaged but also adapting effective strategies to achieve the intended goal.

To realize effective persuasive dialogue systems, recent years have witnessed several promising persuasion models. For example, Shi et al. (2021) utilizes reinforcement learning methods to avoid repetition and inconsistency in the persuader's responses; Chen et al. (2022) integrates factual information and social content into persuasive dialogue systems; Samad et al. (2022) developed an empathetic persuasive dialogue system, etc. Although these models have achieved remarkable success, they primarily focus on designing improved algorithms from a semantic perspective to generate or match responses, rather than efficiently capitalizing on the vital role of conversational strategies.

Skilled persuaders may use high-level strategies to attract and guide the users to achieve the predefined goals. As shown in Figure 1, when continuing the persuasive conversation at the fourth turn, the agent may improve the effectiveness of the conversation by first using the phrase "I understand" to

address the user's emotions. Subsequently, utilizing a *logical-appeal* strategy before *foot-in-door* is more likely to result in the person accepting a donation proposal, rather than solely relying on a *foot-in-door* strategy by asking "think about donating a little bit of 50 cents". The conversation strategy and utterance play different yet complementary roles in persuasion modeling. The strategy abstracts the conversations in a more direct and global manner. Successful persuasions are usually determined by effective transition patterns among different strategies. The utterances are the specific outcomes leveraged to interact with the users. They contain more detailed information that is not covered in the strategies. As a result, it is essential to consider both conversation strategies and utterances jointly to promote better persuasion models.

Inspired by the above analysis, we propose to build a persuasion dialogue model by simultaneously matching strategies and semantic. Specifically, we propose using an auto-regressive decoder to generate next strategies sequentially. Then we propose a new framework that combines both semantic and strategy matching scores as the final score for response candidates. We evaluate our proposed methods and state-of-the-art baselines on a small publicly available dataset (Wang et al., 2019) and a newly collected dataset that are 30 times larger. Experimental results indicate that our proposed approach significantly outperforms all baselines in terms of Recall@1, by 5% on the small dataset and 37% on the large dataset.

The main contributions of this paper are summarized as follows: (1) We propose to build persuasion models by jointly considering the conversation strategies and semantic, which, to our best knowledge, is the first time in this domain. (2) To achieve the above goal, we propose an auto-regressive model to generate next conversation strategy from previous strategies and utterances. (3) We collected a large dataset from a real-world application and performed extensive offline experiments and an online evaluation, which demonstrate the effective use of our model in persuasion.

## 2  Problem Formulation

This paper investigates the persuasive dialogue between two interlocutors, referred to as the *persuader* or "+" party and the *persuadee* or "−" party. The task is formulated as a retrieval-based problem, in which the model should select an appropriate

response for the persuader in order to continue the conversation and try to achieve a persuasion goal. Formally, we define a session of persuasive dialogue as a pair $(\mathcal{D}, R)$. $\mathcal{D}$ is the dialogue content, and $R$ is a binary *return* indicating whether the persuasion is ultimately successful ($R = 1$) or not ($R = 0$). The signal $R$ is only available during the training stage. The dialogue content is given by $\mathcal{D} = \{u_1^+, u_1^-, \ldots, u_N^+, u_N^-\}$, where $u_t^+$ and $u_t^-$ are the persuader's and persuadee's utterances at the $t$-th turn, respectively. Each $u_t$ contains $l_t$ tokens.

There are $K$ atomic strategies that can be used by the persuader. Note that the persuader can use multiple strategies in a single utterance. In this paper, we train multi-label classifiers from the data with strategy labels (see Section 4.1) and use them to automatically label all utterances. We further denote the strategy set at the $t$-th turn as a binary vector $\mathcal{A}_t \in \mathbb{R}^K$, where $\mathcal{A}_{t,i} \in \{0, 1\}$ indicating whether the $i$-th strategy is used or not. Based on the above notations, we learn a matching model $f$ with the following input and output:

**Input**: a dialogue context $\{u_1^+, u_1^-, ..., u_t^+, u_t^-\}$ associated with a strategy set $\{\mathcal{A}_1, ..., \mathcal{A}_t\}$, and a response-strategy pair $(u_c^+, \mathcal{A}_c)$

**Output**: a matching score between $(0, 1)$.

Based on $f$, we select the next response with the largest matching score.

## 3  Our Approach

Effective persuasive conversation requires a next utterance that is fluent, appealing and presents strategies that are successful in influencing users to achieve the persuasion goal. As shown in Figure 2, we propose a new framework called SARA (**S**emantic **A**nd st**R**ategy m**A**tching) that assesses the quality of response candidates from both **utterance** and **strategy** perspectives. For the utterances, we calculate the matching score between the candidate $u_c^+$ and context $\mathcal{C}$ by a fine-tuned BERT and two auxiliary tasks. As for the strategies, we design an auto-regressive model to generate the next strategies, which are expected to capture the strategy transition patterns, and estimate the matching score between the strategies of candidate $\mathcal{A}_c$ and the generated strategies. Finally, we aggregate the two scores as the final score for ranking candidates and select the most appropriate response. In the following, we introduce our model more in detail.

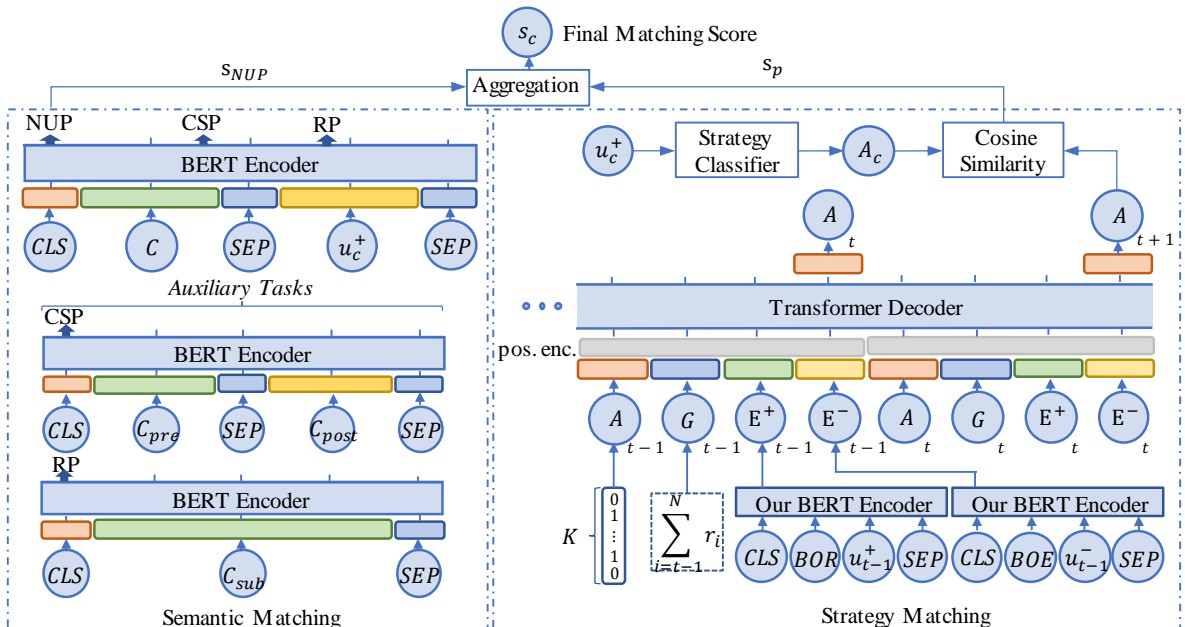

Figure 2: Overview of our SARA framework. At the $t$-th turn, $u_t^+$ and $u_t^-$ are persuader's and persuadee's utterances, $E_t^+$ and $E_t^-$ are the semantic representation of $u_t^+$ and $u_t^-$, $\mathcal{A}_t$ is the used strategies, $G_t$ is the expect gain.

## 3.1 Semantic Matching

Recent retrieval-based dialogue systems commonly use BERT for the task of choosing a response that fits naturally with the preceding context. In SARA, we also employ BERT to measure the semantic similarity between a dialogue context and a response candidate. Before fine-tuning the model, we adapt the BERT model parameters to address potential domain differences by implementing the post-training proposed by Han et al. (2021) over our datasets.

The key fine-tuning objective, as shown in Figure 2, is to determine the extent to which a candidate response $u_c^+$ matches the semantic content of the conversation context $\mathcal{C}$. We refer to this task as **Next Utterance Prediction (NUP)** task. Following the design of BERT, we construct the input sequence $x_{\text{NUP}}$ as:

$$x_{\text{NUP}} = [\texttt{[CLS]}; \mathcal{C}; \texttt{[SEP]}; u_c^+; \texttt{[SEP]}],$$

$\texttt{[CLS]}$ and $\texttt{[SEP]}$ are two special tokens of BERT, and [;] is concatenation operation.

Different from previous works, we add special tokens $\texttt{[BOR]}$ and $\texttt{[BOE]}$ to indicate the role of the utterance in the context $\mathcal{C}$. These tokens are placed at the beginning of persuadeR's and persuadeE's utterances, respectively, before concatenating them into a long sequence. We do the same thing for concatenating multiple utterances of context in the remaining part of this paper. Next, the sequence is fed into BERT, and the $\texttt{[CLS]}$ token, i.e., $\mathbf{E}_{\text{NUP}}$, is

used to predict the matching score, as follows:

$$s_{\text{NUP}} = \sigma\left(\mathbf{W}_{\text{NUP}}^\top \mathbf{E}_{\text{NUP}} + b_{\text{NUP}}\right), \quad (1)$$

where $\mathbf{W}_{\text{NUP}} \in \mathbb{R}^{768 \times 1}$ and $b_{\text{NUP}} \in \mathbb{R}$ are parameters. $\sigma(\cdot)$ is the sigmoid activation function.

To enhance the model's capacity for detecting long-term dependencies in dialogues, we propose two auxiliary tasks: Context Segment Prediction (CSP) and Return Prediction (RP). For CSP, we divide the conversation context $\mathcal{C}$ into two segments at the randomly selected turn $t$. The resulting segments are $\mathcal{C}_{\text{pre}} = \{u_1^+, u_1^-, \ldots, u_t^+, u_t^-\}$ and $\mathcal{C}_{\text{post}} = \{u_{t+1}^+, u_{t+1}^-, \ldots, u_{n-1}^+, u_{n-1}^-\}$. We then construct the input $x_{\text{CSP}}$ as follows:

$$x_{\text{CSP}} = [\texttt{[CLS]}; \mathcal{C}_{\text{pre}}; \texttt{[SEP]}; \mathcal{C}_{\text{post}}; \texttt{[SEP]}].$$

Negative samples were randomly selected from other dialogues' context segments. The BERT model uses a linear layer with a sigmoid function that takes the representation of the $\texttt{[CLS]}$ token as input to predict whether two segments belong to a consecutive context. For RP, the model predicts the return $R$, *i.e.*, whether the persuasion goal was achieved or not, using a partial dialogue context as input. We randomly select a subset from $\mathcal{C}$, denoted as $\mathcal{C}_{\text{sub}} = \{u_{t_1}^+, u_{t_1}^-, \ldots, u_{t_2}^+, u_{t_2}^-\}$. Then, we construct the input $x_{\text{RP}}$ as:

$$x_{\text{RP}} = [\texttt{[CLS]}; \mathcal{C}_{\text{sub}}; \texttt{[SEP]}].$$

The BERT model is trained to predict the return $R$ based on the representation of the [CLS] token.

**Training**    The three fine-tuning tasks are based on binary classification and learned jointly using multi-task learning. The loss function for the three tasks is defined as:

$$\mathcal{L}_1 = \mathcal{L}_{\text{NUP}} + \gamma_1 \cdot \mathcal{L}_{\text{CSP}} + \gamma_2 \cdot \mathcal{L}_{\text{RP}}, \quad (2)$$

where $\gamma_1$ and $\gamma_2$ are hyper-parameters to control the impacts of two auxiliary tasks. Cross-entropy loss is used to implement $\mathcal{L}_{\text{NUP}}$, $\mathcal{L}_{\text{CSP}}$, and $\mathcal{L}_{\text{RP}}$.

## 3.2 Strategy Matching

In persuasive dialogue, the primary goal is to persuade the other interlocutor, in addition to continuing a dialogue. People often employ a series of persuasion strategies. For instance, in the case of persuading others to donate, one can first describe the situation of catastrophe victims to elicit sympathy, inquire about their career, and finally suggest a suitable donation amount. We propose a sequential strategy generation method, inspired by a recent study on offline reinforcement learning (Chen et al., 2021), to model such a persuasion process and predict future strategies in an auto-regressive manner using the previous turns as input.

**Turn Representation**    To model the persuasion process sequentially, we view a dialogue session as several turns and regard each turn as a basic representation unit. Within the $t$-th turn, we consider four elements: the persuasion strategy set, $\mathcal{A}_t$, the expected gain, $G_t$, and representations of the utterances $u_t^+$ and $u_t^-$, denoted as $\mathbf{E}_t^+$ and $\mathbf{E}_t^-$.

(1) Strategy Vector: $\mathcal{A}_t \in \mathbb{R}^K$ is a binary vector, where the size $K$ is the number of atomic strategies.

(2) Expected Gain: The return, $R$, is divided into rewards from each turn, from which we calculate the expected gain. At each turn $t$, we calculate the reward as $r_t = R/N$, where $N$ is the total number of turns. We then calculate the expected gain $G_t$ by summing the rewards from the current turn up to the final turn: $G_t = \sum_{i=t}^{N} r_i$. [1]

(3) Utterances: We represent the text $u_t^+$ and $u_t^-$ by our BERT model with the input format as:

$$x_t^+ = [[\text{CLS}]; [\text{BOR}]; u_t^+; [\text{SEP}]],$$
$$x_t^- = [[\text{CLS}]; [\text{BOE}]; u_t^-; [\text{SEP}]].$$

---

[1] Experimental results show that this method is the most stable for the training process. A more efficient method for distributing the return can be explored in future work.

The representations $\mathbf{E}_t^+$ and $\mathbf{E}_t^-$ are obtained by using the outputs of [CLS] tokens. To save computation cost, the parameters are shared with our fine-tuned BERT in Section 3.1.

As yet, the four elements have been represented as vectors with different dimensions. We apply a group of linear transformations to align them into the same space. For example, for the utterance representations $\mathbf{E}_t$, we apply $T : \mathbb{R}^{768} \to \mathbb{R}^d$ to compute a $d$-dimensional vector. Finally, we obtain a quadruple $(\mathbf{A}_t, \mathbf{G}_t, \mathbf{E}_t^+, \mathbf{E}_t^-)$ that corresponds to the $t$-th turn, where all elements have the same $d$ dimension (to avoid redundancy, we do not use additional symbols for utterance representations after transformation).

**Auto-regressive Strategy Prediction**    After representing dialogues as turn sequences, we propose to learn the persuasion process by predicting the strategies at each turn in an auto-regressive manner based on a Transformer decoder (Vaswani et al., 2017). To elaborate, we structure all components in a sequence for a $N$ turn dialogue: $\tau_t = [\mathbf{A}_t, \mathbf{G}_t, \mathbf{E}_t^+, \mathbf{E}_t^-]$, and $\mathcal{T}_N = [\tau_1, \ldots, \tau_N]$. where $\mathcal{T}_N \in \mathbb{R}^{4N \times d}$. To enable sequential learning in an auto-regressive manner, we add position embeddings upon the quadruple $\tau_t$ to indicate the turn's index and feed $\mathcal{T}_t$ as input to a Transformer decoder. The generation process's objective is:

$$\arg\max_{\Theta} \prod_{t=1}^{N} p(\mathcal{A}_{t+1} | \mathcal{T}_t; \Theta), \quad (3)$$

where $p$ is the generation probability, and $\Theta$ denotes the Transformer decoder's parameters.

**Training**    We implement the objective by the following network:

$$\hat{\mathcal{A}}_{t+1} = \text{sigmoid}(\mathbf{W}\mathbf{H}_{E_t^-} + \mathbf{b}), \quad (4)$$

where $\mathbf{H}_{E_t^-}$ is the hidden state of $\mathbf{E}_t^-$. $\mathbf{W} \in \mathbb{R}^{d \times K}$ and $\mathbf{b} \in \mathbb{R}^K$ are learnable parameters. $\hat{\mathcal{A}}_{t+1} \in \mathbb{R}^K$ is a distribution vector within the range of (0, 1), indicating the probabilities of selecting various strategies at the turn $t + 1$.

Then the learning objective is attained by optimizing binary cross-entropy loss as follows:

$$\mathcal{L}_2 = \sum_{t=1}^{N} \sum_{j=1}^{K} [-\mathcal{A}_{tj} \cdot \log \hat{\mathcal{A}}_{tj}$$
$$+ (1 - \mathcal{A}_{tj}) \cdot \log(1 - \hat{\mathcal{A}}_{tj})], \quad (5)$$

Table 1: Statistics of two persuasive dialogue datasets.

| Statistics | PersuasionForGood | DebtRiskAlert |
|---|---|---|
| Dialogue sessions | 1,017 | 30,000 |
| Annotated dialogues | 300 | 4,000 |
| Atomic strategies | 27 | 83 |
| Turns per dialogue | 10.43 | 4.98 |
| Words per utterance | 19.36 | 23.11 |
| Strategies per turn | 1.82 | 2.38 |
| Persuaded | 545 (53.6%) | 17,947 (59.8%) |
| Not persuaded | 472 (46.4%) | 12,053 (40.2%) |

**Matching** In inference phase, we use the cosine similarity between the predicted $\hat{\mathcal{A}}_{t+1}$ and the candidate strategy set $\mathcal{A}_c$ of $u_c^+$ to calculate the persuasion strategy matching score $s_p$ as follows:

$$s_p = \frac{\mathcal{A}_c^\top \hat{\mathcal{A}}_{t+1}}{\|\mathcal{A}_c\|_2 \cdot \|\hat{\mathcal{A}}_{t+1}\|_2}. \qquad (6)$$

It is worth noting that we cannot obtain the final persuasion return $R$ in advance during inference. Therefore, we set $N$ to the average number of turns per session and $R = 1$, to encourage the model to generate the corresponding persuasion strategies and pursue the persuasion goal.

### 3.3 Fusion for Ranking

The final score is computed based on how good a response candidate is as the next utterance for the given context and how well it matches the predicted strategies. Concretely, we linearly combine $s_{\text{NUP}}$ and $s_p$ together for a joint ranking score:

$$s = (1 - \delta) \cdot s_{\text{NUP}} + \delta \cdot s_p, \qquad (7)$$

where $\delta$ is a hyper-parameter to balance the scores. Such a fusion is widely used and can easily interpret the importance of two factors.

Although we attempt to optimize both $\mathcal{L}_1$ and $\mathcal{L}_2$ jointly, the results are unsatisfactory and exhibit instability during the training process. Therefore, we separately optimize the semantic matching loss $\mathcal{L}_1$ and strategy matching loss $\mathcal{L}_2$, and empirically set the optimal $\delta$ by using validation set finally.

## 4 Data and Evaluation

### 4.1 Datasets from Applications

To better evaluate approaches in real applications, we built an additional large dataset called DebtRiskAlert, which is 30 times larger than the public dataset PersuasionForGood, as shown in Table 1.

**PersuasionForGood** This dataset (Wang et al., 2019) simulates a persuasive scenario in which a group of individuals persuades others to donate to the charity organization named *Save the Children*. The sessions are labeled as 1, *i.e.*, $R = 1$, if the persuadee donated or otherwise 0. Persuaders can use 27 atomic strategies, *i.e.*, $K = 27$, such as *self-modeling* and *emotion-appeal*, and *confirm-donation*. As there are only 29.5% of dialogues are labeled with persuasion strategy, we fine-tune a RoBERTa model (Liu et al., 2019) on these annotated data as a classifier, and use it to obtain predicted strategies for the unlabeled dialogues. This approach achieves an accuracy of 72.9%.

**DebtRiskAlert** We collect another dataset from a real-world persuasion scenario to support this study on a commercial platform, where skilled debt reminders alert users with overdue debts by calls and try to persuade them to repay their debt so as to reduce financial risk. We collect 30,000 dialogues, which are labeled as 1, *i.e.*, $R = 1$, if the users finally make repayments within several days or otherwise 0. A total of 83 strategies, *i.e.*, $K = 83$, are used by the persuaders, such as *credit-protection*, *partial-repay*, and *amount-notification*. 4,000 dialogues are manually annotated with strategy labels. Similar to the PersuasionForFood dataset, for unlabeled data, we obtain predicted labels by a trained classifier with an accuracy of 85.0%.

Comparing the two datasets (See Table 1), we find that DebtRiskAlert has longer utterances but shorter sessions than PersuadeForGood. This indicates that DebtRiskAlert call receivers have less patience, although the success rate in DebtRiskAlert is a bit higher than PersuadeForGood. Each turn in the two datasets has 1.8 and 2.4 strategies on average, which indicates that applying multiple strategies is common in persuasive dialogue.

### 4.2 Evaluation

Retrieval-based dialogue systems are often evaluated as follows (Han et al., 2021; Yuan et al., 2019). First, given a dialogue context, the original response from humans is regarded as a positive utterance. Then, the randomly sampled $M$ utterances ($M = 99$ in our paper) from other dialogue sessions are regarded as negative candidates, although some of them may be false negative. Next, all the compared methods rank the $M + 1$ candidates, and the performance is evaluated by the rank of the ground-truth response. Recall at $k$ (R@$k$, $k$={1,2,5}) is used as the evaluation metric.

Different from an open-domain chitchat system,

Table 2: Comparing dialogue models with our approach on two benchmarks. The best results are in **bold**, and † indicates significant improvement over all baselines with $p$-value $< 0.05$.

|  | PersuasionForGood | | | DebtRiskAlert | | |
|---|---|---|---|---|---|---|
|  | R@1 | R@2 | R@5 | R@1 | R@2 | R@5 |
| SMN | 4.5 | 8.4 | 16.7 | 20.5 | 34.1 | 56.8 |
| MSN | 11.7 | 17.5 | 29.3 | 29.3 | 42.1 | 61.4 |
| BERT | 31.7 | 46.5 | 65.5 | 32.3 | 49.1 | 75.6 |
| SA-BERT | 29.7 | 44.7 | 65.4 | 33.6 | 50.1 | 78.2 |
| BERT-FP | 37.8 | 53.1 | 71.3 | 47.9 | 68.2 | 87.0 |
| + CQL | 38.1 | 53.3 | 71.2 | 47.3 | 68.1 | 87.0 |
| + BCQ | 38.1 | 53.3 | 71.3 | 61.2 | 72.7 | 87.3 |
| **SARA (ours)** | **39.8**† | **53.6** | **71.7** | **65.8**† | **75.6**† | **89.2**† |
| w/o Strategy | 38.3 | 53.3 | 71.7 | 50.6 | 70.1 | 88.6 |

Table 3: Comparing strategy selection models with our approach on two benchmarks. ACC denotes accuracy when all predicted labels exactly match with ground-truth. The best results are in **bold**.

|  | PersuasionForGood | | DebtRiskAlert | |
|---|---|---|---|---|
|  | ACC | F1-score | ACC | F1-score |
| BCQ | 11.0 | 22.4 | 41.9 | 44.8 |
| CQL | 19.3 | 31.5 | 21.5 | 12.2 |
| **SARA Strategy** | **20.9** | **40.1** | **45.8** | **52.8** |
| w/o $G$ | 20.8 | 39.6 | **45.8** | 52.2 |
| w/o $E^+$ | 20.5 | 33.9 | 39.5 | 47.0 |
| w/o $E^-$ | **20.9** | 34.1 | 34.6 | 32.6 |
| w/o $E^+$ & $E^-$ | 20.0 | 31.3 | 30.3 | 32.1 |

a persuasive dialogue system in our real application requires the model to select appropriate strategies first and then sample some human reviewed utterances of the strategies to compose a final utterance, in order to control data quality and non-compliance risk. Thus, we randomly sample $M$ utterances from other strategies than the ground-truth strategy as negative samples for evaluating persuasive systems. Moreover, this way can also reduce the number of negatives that are actually appropriate responses. We manually label 400 negative samples ranked at the top one, based on the PersuasionForGood dataset, and find that sampling negative utterances results in a 12% false negative rate, while sampling utterances from negative strategies yields only 7%. Please note our strategies are automatically classified and thus we cannot guarantee the utterances with negative strategy labels are 100% negative.

## 5 Experiments

We conduct experiments to compare our model with two categories of retrieval-based dialogue models, and compare our strategy matching module with strategy selection baselines. Moreover, we do analyses to gain in-depth insights. We also conduct an online evaluation by deploying our SARA model to real users. Implementation details of SARA are described in Section A.1 of Appendix.

### 5.1 Baselines

**Interaction-based Models** The paradigm of interaction-based retrieval dialogue methods is: 1) individually encoding each utterance in the dialogue context and the candidate response; 2) fully interacting the representation of each utterance with that of the candidate response; 3) aggregating

the interaction information for ranking. We choose two typical methods, SMN (Wu et al., 2017) and MSN (Yuan et al., 2019), as baselines.

**PLM-based Models** The manner of pre-trained language model (PLM)-based methods is: 1) choosing a pre-trained language model (*e.g.*, BERT) as the backbone; 2) concatenating the context and candidate response into a long sequence as the input; and 3) using the output representation to compute the ranking score by a neural network layer. We choose BERT (Devlin et al., 2019), SA-BERT (Gu et al., 2020), and BERT-FP (Han et al., 2021) as PLM-based baselines, where BERT-FP is the state-of-the-art method in the response selection task.

**Strategy Selection Models** Offline Reinforcement Learning can be used to sequentially predict strategies, and thus we choose Conservative Q-Learning (CQL) (Kumar et al., 2020) and Batch Constrained Q-learning (BCQ) (Fujimoto et al., 2019) as baselines. Their implementation details are provided in Section A.2 of Appendix.

### 5.2 Comparison with Dialogue Models

We compare our SARA model with the dialogue baselines and show results in Table 2. SARA performs the best in all metrics over the two datasets. The improvements are also statistically significant in terms of R@1, the major metric for the target applications. Due to the small size of the PersuasionForGood dataset, non-pretrained methods, *i.e.*, SMN and MSN, cannot be fully optimized and result in low performance. Our strategy generative model may also not be fully optimized, but it still significantly achieves 5% relative improvement in R@1. On the larger DebtRiskAlert dataset, our method has a relative improvement of 37% in R@1 and the improvements in R@2 and R@5 (36% and 11%) are also significant. These results

demonstrate the best effectiveness of our proposed method, particularly when data is plentiful and the strategy generative model can be fully optimized.

Table 2 also shows that all proposed ideas have positive contributions. Using the two auxiliary tasks, *i.e.*, Context Segment Prediction and Return Prediction, to fine-tune BERT can improve the baseline slightly from 37.8% to 38.3% in R@1 on the small dataset and dramatically from 47.9% to 50.6% on the large dataset. Without strategy matching the performance drops by 1.5% and 15.2% in R@1 on the small dataset and the large dataset.

The previous dialogue models did not consider persuasive strategies. To be fair, we also combine the best baseline model BERT-FP with two baseline strategy selection methods, i.e., BCQ and CQL. The results are presented in Table 2. It indicates that BERT-FP+BCQ is better than BERT-FP+CQL, but our SARA method still significantly outperforms both of them.

## 5.3 Comparison of Strategy Models

We evaluate how well our proposed method, two offline reinforcement learning baselines, and several variants of ours, on predicting the next persuasion strategies. We use accuracy for multi-label classification and average F1-score for different labels as metrics. Table 3 shows the results.

As shown in the table, our proposed method is the best among all methods. Our method improves the best performance of two baselines by 8% in accuracy and 27% in F1-score on the PersuasionForGood data and by 9% in accuracy and 17% in F1-score on the DebtRiskAlert data. This indicates that our proposed strategy generation is the most effective in predicting the next strategies. When we remove the expected gain, the performance slightly drops; whereas when removing $\mathbf{E}_t^+$, $\mathbf{E}_t^-$, or both, the F1-score dramatically decreases. This indicates that predicting the next strategies heavily relies on previous utterances of persuaders and persuadees. Previous strategies are the main contributors as using them and gains (w/o $\mathbf{E}_t^+$ & $\mathbf{E}_t^-$) can still achieve 31.3 and 32.1 in F1-score on two datasets.

## 5.4 Effect of the Fusion Weight

Figure 3 illustrates the effect of the fusion weight $\delta$ on the performance of our proposed method. When $\delta = 0$, it equals to semantic matching only; whereas it equals to strategy matching only when $\delta = 1$. On the DebtRiskAlert dataset, the strategy matching module has significantly outperformed

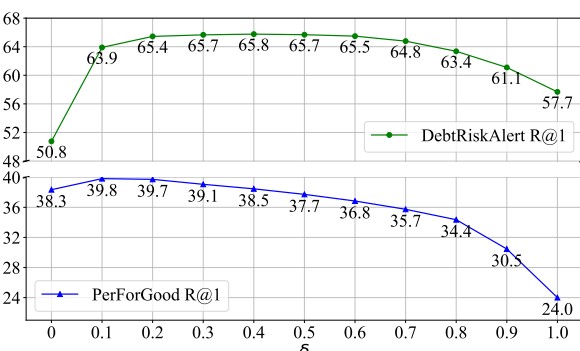

Figure 3: Impact of hyper-parameter $\delta$ on the performance of the proposed method.

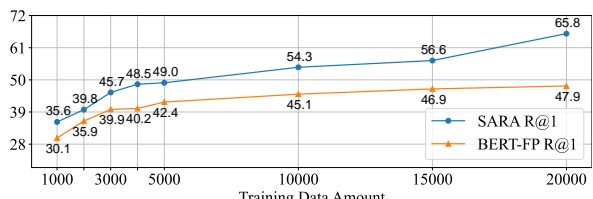

Figure 4: Performance according to the amount of training data in DebtRiskAlert.

the semantic matching module. When they are combined, we can obtain significantly better results when $\delta$ is in the wide range from 0.2 to 0.6. It indicates that combining the two modules stably leads to a significant improvement in performance. Over the small dataset, although the strategy matching module cannot be fully optimized and get lower performance than the semantic matching module, the fused results are still better than semantic matching alone when $\delta$ is from 0.1 to 0.2. Again, it indicates that our SARA framework is consistently effective.

## 5.5 Impact of Training Data Scale

We investigate whether the performance difference is caused by the scale of training data. Thus, we randomly sample 1,000 sessions from the DebtRiskAlert dataset, which is the same scale as the PersuadeForGood dataset, to train our model and then increase the scale of training data little by little. We draw their performance in Figure 4. When using 1,000 sessions, the performance drops to 35.6 in R@1, which is even lower than that on the PersuadeForGood. With increasing training data, the performance of our model continues to improve. This indicates that we do need large-scale data to train our models and draw convincing conclusions. In addition, although the performance of the BERT-FP baseline improves with more training data as well, the increased gap between the two curves indicates that more training data benefit strategy matching more than semantic matching.

## 5.6 Online Evaluation

To deploy the proposed SARA model on the debt risk alert platform, we make several additional efforts to ensure quality, safety and user experience. First, we acquired 388 human confirmed high-quality utterances, which is much fewer than the number of candidate utterances in offline evaluation, covering all 83 strategies. Then we distilled the SARA model to a smaller size to comply with latency requirements. To ensure speech quality, we invited skilled human speakers to convert those 388 textual utterances into spoken utterances. Finally, we use the distilled SARA model to select appropriate responses and concatenate their corresponding spoken utterances when talking to real users.

Among the 9,396 calls that SARA made within five days, 2,204 users repay their debts on the day they receive the call, resulting in a repayment rate of 23.5%. During the same period, the best online model based on semantic matching and heuristic rules achieves a repayment rate of 21.9%, and the repayment rate for those who do not receive any call is 15.8%. This indicates that our SARA model improves the effectiveness of achieving the ultimate goal of persuasion in real-world applications.

## 6 Related Work

**Retrieval-based Dialogue Systems** aim to select proper responses from a large-scale response repository by estimating the matching degree between context and response (Wu et al., 2017; Yuan et al., 2019). Pre-trained methods have become the dominant approach (Gu et al., 2020; Xu et al., 2021; Han et al., 2021). Directly fine-tuning the pre-trained model as a dialogue model is less effective because of a domain gap. A common practice is to further train the model with the language modeling objectives using texts from the target domain to reduce the negative impact (Han et al., 2021; Gu et al., 2020). As a preliminary study of modeling persuasion strategy sequentially in persuasive dialogue, we follow the retrieval-based paradigm. In addition to being easier to evaluate, it is more controllable and thus can respond ethically in real applications.

**Persuasive Dialogue Systems** aim to influence the views, attitudes, or behaviors of people through conversation. Compared to task-oriented and chitchat-oriented dialogue, identifying and applying persuasion strategies are unique and essential in developing persuasive dialogue systems (Carlile et al., 2018; Wang et al., 2019; Tian et al., 2020;

Jain and Srivastava, 2021). Some work (Wu et al., 2021) assumes that persuasion strategies are hidden in the persuader's utterances, and thus a separate conversation model is learned from the persuader's content, which is reinforced by the feedback from a model learned from the other person. Some works (Shi et al., 2021; Samad et al., 2022) build a persuasion strategy classifier and add the loss of matching the strategies of generated and ground-truth responses during the training, but they do not explicitly use strategies in inference. Some work (Chen et al., 2022) assumes that the proper strategy is given and investigate how to take the strategy as a condition to generate better response. Different from the previous works, to our best knowledge, we are the first to explicitly predict the proper strategies by the whole sequence of previous strategies, utterances from the persuader and the other person, and explicitly integrate the predicted strategies in scoring the best response.

**Offline Reinforcement Learning (RL) Approaches** provide an offline learning paradigm for constructing sequential decision-making engines based on the fixed dataset without online environment interaction. Recently, offline RL approaches have been used in task-oriented dialogue systems, which were used to optimize the policy of response selection to achieve user goals (Jang et al., 2021; Wen et al., 2017; Peng et al., 2018). Since some representative offline RL algorithms, such as Decision Transformer (Chen et al., 2021) can conduct sequential action selection conditioned on a preset target, they provide a promising way to model the sequential strategy generation process and inspire our ideas to improve persuasive dialogue.

## 7 Conclusion

In this paper, we propose a new framework called SARA that jointly considers semantic and strategy matching in persuasive dialogue. For strategy matching, we propose using an auto-regressive generator to sequentially predict the next appropriate strategies based on all previous utterances, strategies, and expected gain to return. We also collect a new large-scale dataset from a real application. Experimental results on the dataset and a small public dataset indicate that integrating strategy matching can significantly improve the state-of-the-art dialogue baselines in terms of R@1. Our proposed strategy generation model performs the best among strategy selection baselines. Online evaluation also

indicates the effectiveness of our model in achieving the ultimate goal of persuasion. The big impact of training data scale suggests using enough data is necessary in future work. We plan to improve our persuasive dialogue systems by supervised data, *e.g.*differentiating high-quality training data from others by the average return of a caller.

## Acknowledgements

This work is supported by the Fundamental Research Funds for the Central Universities, and the Research Funds of Renmin University of China (21XNLG28), and National Natural Science Foundation of China (No. 62276268).

## Limitations

Due to concerns about the compliance and quality of online experiments, it is difficult to directly evaluate the persuasion outcomes of different methods online. However, as a preliminary exploration of supervised retrieval-based persuasive dialogue systems, there is still a gap between the current persuasive dialogue system and human generated supervised data. Thus, it is reasonable to evaluate the persuasive dialogue system with real human responses. Considering user privacy, we cannot directly release the DebtRiskAlert dataset. However, it is not difficult to collect and construct similar persuasive dialogue datasets when researchers cooperate with the industry.

## Ethical Statement

The study of persuasive dialogue systems is still in its early stages. As the system is designed to achieve some given purposes, such as changing the user's attitude, it should be carefully checked to avoid providing offensive utterances. In this study, we formulate the research problem in a retrieval-based setting. Therefore, we have all utterance candidates manually checked and ensure that all of them are proper for an online dialogue. Besides, in our collected dataset (DebtRiskAlert), we have removed all user-related information to keep user privacy. It is worth noting that, though our system can be applied to various scenarios, it should not be used to replace humans directly. All applications should be performed under human supervision.

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

## A  Appendix

### A.1  Implementation Details of SARA

We implement our model by PyTorch (Paszke et al., 2019). The maximum length of the input sequence is set to 256. If the length of the input sequence exceeds the maximum length, we remove tokens from the shorter one of the context and the candidate response. $\gamma_1$ and $\gamma_2$ are both set to 0.1. The BERT model for dialogue modeling is trained for 10 epochs with a batch size of 96. Adam optimizer (Kingma and Ba, 2015) is applied with a learning rate of 1e-5. The strategy decision model is implemented based on Decision Transformer.[2] The maximum length of the input sequence is set as 120. The representation dimension $d$ is set to 128. The number of hidden layers and heads is set to 12 and 8, respectively. The model is also optimized by the Adam optimizer with a learning rate set of 1e-5. The training epochs is 50 with a batch size of 8. In the inference stage, we set the return and turns to 1 and 10, respectively. Following previous work (Wang et al., 2019), we perform five-fold cross-validations on the PersuasionForGood dataset and use the average scores of each fold to compare different models. For the DebtRiskAlert dataset, it is divided into 20k, 5k, and 5k dialogues as training, validation, and testing sets, respectively. We train our model on 2 v100 GPUs, and it takes about 16 hours to run all the experiments on both datasets.

### A.2  Details of Offline RL Baselines

When considering the offline reinforcement learning methods CQL and BCQ as baselines, the key

---

[2]https://github.com/kzl/decision-transformer

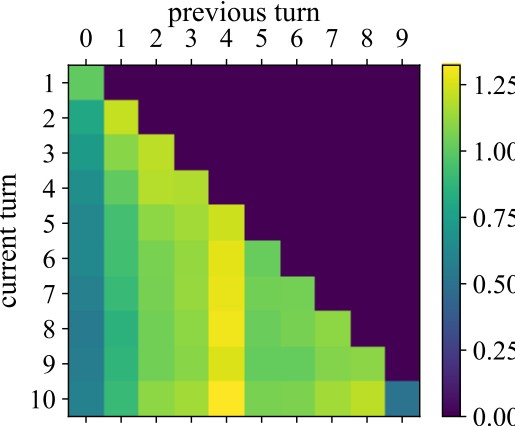

Figure 5: Average normalized attention weights of different turns over the entire PersuasionForGood dataset.

issue lies in designing the state, action, and reward. When predicting the action for the $t + 1$ turn, we take the following design into account:

- **Action**: We use the binary strategy vector $\mathcal{A}_{t+1}$ as the action at the $t + 1$ turn;

- **State**: We consider using persuadee's utterances $u_t^-$, persuader's utterance $u_t^+$ and corresponding persuasion strategies $\mathcal{A}_t$ as states. So we concatenate the feature representations $E_t^+$, $E_t^-$ and $\mathcal{A}_t$ as input;

- **Reward**: We calculate the reward as $r_t = R/N$, where $R$ represents the return and $N$ denotes the total number of turns.

It is worth mentioning that, based on this design, the offline RL baselines align with the proposed strategy selection model, as they also integrate semantic and strategy information as input.

### A.3  Attention Weight Analysis of Persuasion

In Figure 5, we normalize the attention weights over different turns based on the PersuasionForGood dataset. The brighter yellow color means higher attention weight while the darker blue color means lower. We find that for any turn, all previous turns are somehow useful except for the first turn, which is most likely greeting and mentioning the topic. The most interesting finding is that the fourth-turn is the most important for the later strategy generation. It may be caused by that the persuadee having an obvious attitude on whether he/she wants to donate. The latter strategies are chosen around this key information.