# OpenReview forum: "Joint Semantic and Strategy Matching for Persuasive Dialogue"
_EMNLP/2023/Conference — EMNLP 2023 Findings_

### Official Review · Reviewer_zkSK · 2023-07-31

**Typos Grammar Style And Presentation Improvements:** See above.
**Soundness:** 3

**Excitement:**

2: Mediocre: This paper makes marginal contributions (vs non-contemporaneous work), so I would rather not see it in the conference.

**Missing References:**

See above.

**Paper Topic And Main Contributions:**

Persuasive dialogue aims to persuade users to achieve specifc goals through conversations. While previous models mostly rely on matching utterance semantics and neglect the strategies of a conversation. Therefore, the paper proposes a novel persuasion framework that combines the modeling of conversation semantics and strategies, which consists of a BERT-like module and an auto-regressive predictor that match the semantics and strategies, respectively. Experiments have shown that the model improve some baselines on two datasets.

**Questions For The Authors:**

	1. On the top left in Fig.2, the "CSP" and "RP" appeared. Is it right?
	2. Compared with ChatGPT, how is the performance? Have tried?


**Reasons To Accept:**

	1. The proposed model outperforms some baselines from 2019-2021.
        2. The paper considers a real-world dataset, looking forward to the dataset and code.

**Reasons To Reject:**

	1. The proposed model is not novel enough, it seems combining the work A and work B, which is not meet the high criteria of EMNLP.
	2. The compared baselines are too old, which comes from 2019, 2020, 2021. More new models and more analyses could be better.
	3. There are two another papers which are relevant with strategy prediction."Do You Know My Emotion? Emotion-Aware Strategy Recognition towards a Persuasive Dialogue System",  "Understanding User Resistance Strategies in Persuasive Conversations"


**Reproducibility:**

2: Would be hard pressed to reproduce the results. The contribution depends on data that are simply not available outside the author's institution or consortium; not enough details are provided.

**Reviewer Confidence:**

3: Pretty sure, but there's a chance I missed something. Although I have a good feel for this area in general, I did not carefully check the paper's details, e.g., the math, experimental design, or novelty.

---

> ### Author Rebuttal · Authors · 2023-08-29
>
> We would like to thank the reviewer for the constructive feedback and provide the following answers to the concerns.
>
> **Q1: The proposed model is not novel enough...not meet the high criteria of EMNLP**
>
> As Conference on Empirical Methods in Natural Language Processing (EMNLP) emphasizes "Empirical" in its name, being "novel" covers not only a new model architecture but also new empirical ideas that are proved to be a significantly better solution to an important task. We shall also measure whether a method is novel by considering how much impact it makes. Our community has too many published papers today but few of them do work in real applications and widely used. The latter influences the impact of a conference indeed.
>
> **Here we clarify our novelty and contributions again:**
>
> 1.  We address an important but difficult problem. Even for ChatGPT, the persuasive dialogue task is difficult and important because it requires the ability to achieve a goal via a long conversation. It is still a challenging task for our community and it is the core ability of many real applications on automatic calling.
> 2.  Our paper is the first work to propose combining strategy selection with semantic matching, which brings significant improvements in the effectiveness of persuasive dialogue systems;
> 3.  We are the first to discover that the small dataset is not enough to verify the effectiveness of an idea on persuasive dialogue and also provide insights on how many is good enough;
> 4.  In addition to offline evaluation, we made great efforts to push our ideas online. Our proposed method serves thousands of users and makes a real impact.
>
> **Q2: The baseline method is outdated**
>
> Different from generation-based dialogue models, retrieval-based dialogue models have not made much progress in recent years. However retrieval-based dialogue models are still widely used in industry, in particular when the product is required to be trustworthy. To our knowledge, BERT-FP is still one of the best-performing methods. Some follow-up works have shown relatively small improvements over it or the improvements are not applicable to our focused problem. Our focus is on how combining the SOTA semantic models with strategy matching works for persuasive dialogue applications. Therefore, it is appropriate to choose BERT-FP as the baseline. We will add a more detailed explanation in the related work section.
>
> **Q3: The two reference papers the reviewer provided**
>
> We appreciate the advice and will include these two papers in our related work session. However, these two papers do not weaken our contributions:
>
> - In ``Do You Know My Emotion? Emotion-Aware Strategy Recognition Towards a Persuasive Dialogue System``, authors investigate strategy recognition, i.e., predicting which strategies the persuader used in historical dialogues. It is related to our strategy classifier, but our task is to plan appropriate persuasive strategies for the persuader for future dialogues.
> - In ``Understanding User Resistance Strategies in Persuasive Conversations``, authors focus on annotating resistance strategies based on historical dialogues (similar to strategy recognition based on historical dialogues mentioned above). This is related but different from our work to predict appropriate next strategies and select the best response.
>
> **Q4: On the top left in Fig.2, the "CSP" and "RP" appeared**
>
> We intend to show the simultaneous training of three tasks in the larger figure at the top, and then show the details of two auxiliary tasks at the bottom. We will improve the illustration in the revised version.
>
> **Q5: Compare with ChatGPT**
>
> After the rise of ChatGPT, we also try to apply LLMs to our persuasive dialogue scenarios. We attempt to use ChatGPT and fine-tune open-source LLMs on our data, but the experimental results were disappointing. Although LLMs can generate seemingly good responses, some of them contain hallucinations and it is difficult to control LLMs to generate trustworthy replies only. Moreover, it is difficult to evaluate the generation models. We are still doing research on how to improve ChatGPT-like models in our domain and how to effectively consider long-term goals in LLMs.

---

### Official Review · Reviewer_xjnT · 2023-08-05

**Soundness:** 2

**Excitement:**

3: Ambivalent: It has merits (e.g., it reports state-of-the-art results, the idea is nice), but there are key weaknesses (e.g., it describes incremental work), and it can significantly benefit from another round of revision. However, I won't object to accepting it if my co-reviewers champion it.

**Paper Topic And Main Contributions:**

This work proposes a method for persuasive dialogues by using a generator to generate a proper strategy for the current turn based on context semantics. Specifically, at each persuader's turn, the strategy matching module would pick the best strategy candidate and then , based on the strategy, the semantic matching module would pick the best response candidate. Evaluation results show significantly better results with the proposed methods.

**Questions For The Authors:**



1. Why do you need L_RP in semantic matching? What is the purpose of this loss?
2. In Problem Formulation, what is the merit of defining the task as a retrieval-based task? Please correct me if I'm wrong, but my understanding is that this definition assumes there is already a pool of candidate responses available. Does it reflect realistic scenarios?

**Reasons To Accept:**

Main reasons to accept:

1. The proposed model combines both strategy and semantic into account for persuation. The application of reinforcement learning is an interesting use in this study.
2. The result looks significantly better than baselines on both datasets.

**Reasons To Reject:**

Main reasons to reject:

1. Paper is a little hard to follow.
     a.  In abstract, it claims that the method improves the state-of-the-art baseline by 5% on a small dataset and 37% on a large real-world dataset in terms of Recall@1. I couldn't find links of these numbers to the results in evaluation sections.
     b. The 2 auxiliary losses in semantic matching training is not very clear to me about how helpful they are.

2. This task is defined as retrieval-based task, which would limit its use to the dialogue systems in general.

**Reproducibility:**

4: Could mostly reproduce the results, but there may be some variation because of sample variance or minor variations in their interpretation of the protocol or method.

**Reviewer Confidence:**

3: Pretty sure, but there's a chance I missed something. Although I have a good feel for this area in general, I did not carefully check the paper's details, e.g., the math, experimental design, or novelty.

---

> ### Author Rebuttal · Authors · 2023-08-29
>
> We would like to thank the reviewer for the constructive feedback and provide the following responses to the concerns.
>
> **Q1: Couldn't find links of these numbers to the results in evaluation sections**
>
> The improvements we report are relative to the baseline BERT-FP: the 5% is calculated by (39.8-37.8)/37.8, and the 37% is calculated by (65.8-47.9)/47.9. We will add relative improvements to Table 2 and revise our paper accordingly.
>
> **Q2: Why we need the two auxiliary losses in semantic matching training**
>
> For our addressed persuasive dialogue problem, not only do we need to select a relevant next utterance, but we also need to pay attention to the long-term goal of persuading the other party. Therefore, we design the Context Segment Prediction task to improve the model's ability to predict the next half-session, which somehow captures the relation between utterances in a higher resolution. The Return Prediction task is used to predict whether the persuadee is persuaded, i.e., the ultimate goal, and thus can help the model know which status it stands on in the progress of persuasion and choose a more suitable response for the current status. As Table 2 shows, with auxiliary tasks (Line SARA w/o strategy) our method outperforms the baseline BERT-FP with the next utterance prediction task only. We will add some explanations in the revised paper.
>
> **Q3: What is the merit of defining the task as a retrieval-based task?**
>
> In our target realistic applications, it require that models give correct, relevant and compliant responses, which current generation-based models can difficult to meet. That is why retrieval-based systems are in demand and widely used. A retrieval-based system can better guarantee the quality of responses and is easy to control. For example, in the DebtRiskAlert scenario, we can review all response candidates to ensure their quality and minimize risks. The retrieval-based systems are also superior in latency to generation-based systems like LLMs.
>
> In addition, our aim is to address the important and challenging problem of how to consider the long-term goal, e.g., persuading the other party, in dialogue systems. This is difficult for both retrieval and generation-based systems. As a good start, it is easier to control variables to investigate the problem in retrieval-based dialogue systems. Once we demonstrate that strategy prediction is effective for persuasive dialogues, we are more confident to explore how to integrate long-term goals into the LLM-based generation models.

---

### Official Review · Reviewer_urL8 · 2023-08-10

**Typos Grammar Style And Presentation Improvements:** 1. A suggestion
**Soundness:** 3

**Excitement:**

3: Ambivalent: It has merits (e.g., it reports state-of-the-art results, the idea is nice), but there are key weaknesses (e.g., it describes incremental work), and it can significantly benefit from another round of revision. However, I won't object to accepting it if my co-reviewers champion it.

**Paper Topic And Main Contributions:**

The paper proposes a novel approach to generate persuasive text that jointly models semantic and strategy matching to create a persuasive dialogue system. The joint modeling of semantics and strategies has not been extensively attempted in the literature previously. The proposed method is evaluated on the Persuasion For Good and Debt Risk Alert datasets, where the reported results surpass previous state-of-the-art results.

The paper is well-written, and the proposed approach appears sound. If the paper gets accepted, I recommend the authors release their dataset and implementation for community use.

**Questions For The Authors:**

1. You mention in line 325 that "we separately optimize the semantic matching loss L1 and strategy matching loss L2". Can you elaborate how did you do this?
2. Why is the drop in performance for the w/o Strategy setting in Table 4 much higher for DebtRiskAlert compared to PersuasionForGood?

**Reasons To Accept:**

1. The paper proposes a novel approach that jointly models strategic and semantic matching. Several baselines, offline and online experiments have been conducted to establish the efficacy of the proposed framework.
2. The authors construct a new dataset that is 30 times larger than the Persuasion For Good to conduct thorough experiments. The dataset would be valuable to the community if released.
3. The paper is well-written and easy to follow.

**Reasons To Reject:**

1. Given the multiple variables introduced in section 3, it'll help improve the readability if the authors include a notations/nomenclature table.
2. Figure 2 can be improved. Currently, it's a bit hard to understand the flow chart properly. It'll be helpful to add an example within the figure to explain its working.

**Reproducibility:**

3: Could reproduce the results with some difficulty. The settings of parameters are underspecified or subjectively determined; the training/evaluation data are not widely available.

**Reviewer Confidence:**

3: Pretty sure, but there's a chance I missed something. Although I have a good feel for this area in general, I did not carefully check the paper's details, e.g., the math, experimental design, or novelty.

---

> ### Author Rebuttal · Authors · 2023-08-29
>
> We would like to express our gratitude to the reviewer for providing valuable feedback. We are encouraged that Reviewer\#urL8 acknowledged our approach and dataset. We have carefully considered the concerns raised and have prepared a response to address them.
>
> **Q1: The variables table and Figure 2**
>
> We will add a table in the revised paper to explain our variables. We will also re-design Figure 2 by including an example for better illustration.
>
> **Q2: How separately optimize the semantic matching loss $L_1$ and strategy matching loss $L_2$**
>
> We first optimize the semantic matching loss $L_1$, then freeze the trained BERT model and optimize the strategy matching loss $L_2$. We have tried to optimize them together, but experimental results show that such a separate training strategy is more stable.
>
> **Q3: Drop in performance for the w/o Strategy setting in Table 4 (Figure 4) much higher for DebtRiskAlert**
>
> We have the same observations as the reviewer does. As Figure 3 shows, the strategy-only model performs much worse than the semantic only model on the small PersuasionForGood dataset; whereas, the strategy model alone can beat the semantic model on the large DebtRiskAlert dataset. These indicate that the strategy model cannot be sufficiently trained on such a small dataset as PersuasionForGood. We need large enough datasets to draw reliable conclusions or discover effective ideas for improving persuasive dialogues.
>
> In addition, compared to the PersuasionForGood dataset, DebtRiskAlert has finer-grained strategy annotations (83 atomic strategies vs. 27 atomic strategies), and the strategy classifier provides more accurate annotations in terms of accuracy (85.0% vs. 72.9%). This makes our strategy selection module more effective on the DebtRiskAlert dataset.
>
> **Q4: Move the related work section after introduction**
>
> We will follow this suggestion and move the related work section after the introduction.

---

### Official Review · Reviewer_pxCx · 2023-08-12

**Soundness:** 3

**Excitement:**

4: Strong: This paper deepens the understanding of some phenomenon or lowers the barriers to an existing research direction.

**Paper Topic And Main Contributions:**

This paper mainly studies persuasive dialogue which aims to persuade interlocutors to achieve specific goals through conversations. This paper proposes a novel dialogue framework that simultaneously matching strategies and semantic. This network utilizes a BERT model to measure the semantic similarity and an auto-regressive decoder to generate next conversation strategies. Extensive experimental results on a benchmark dataset and a self-constructed dataset verify the effectiveness of the proposed framework.

**Reasons To Accept:**

1. This paper is well-written. The motivation is clearly stated, and the contribution and methods are clearly described.
2. This paper proposed a novel framework called SARA to assess the quality of response candidates from both utterance and strategy perspectives.
3. In contrast to prior work, the authors considered the strategy matching degree in response selection, and developed an auto-regressive decoder to generate persuade strategies sequentially. This is a novel attempt in the persuasive dialogue task.
4. The authors constructed a large-scale dataset DebtRiskAlert to facilitate the persuasive dialogue research field. Extensive experimental results on both the public benchmark dataset and self-collected dataset verify the effectiveness of the proposed network in this paper.

**Reasons To Reject:**

1. The R@2 and R@3 results of experiment show that the proposed model is only 0.3-0.4 points higher than SOTA on PersuasionForGood dataset. Meanwhile, the w/o strategy result is the same as the main results on the R@5 metrics, which may be difficult to convince the effectiveness of the proposed method.
2. The collected dataset DebtRiskAlert is not public-available, which hinders the reproducibility of the experiments.
3. The author mentions that the persuader can use multiple strategies in a single utterance. The authors could clarify how they solve the problem of multiple choice of strategies in the method
4. The SOTA model BERT-FP used by the authors is from year 2021. The authors could chose a more up-to-date model for the comparison.

**Reproducibility:**

3: Could reproduce the results with some difficulty. The settings of parameters are underspecified or subjectively determined; the training/evaluation data are not widely available.

**Reviewer Confidence:**

4: Quite sure. I tried to check the important points carefully. It's unlikely, though conceivable, that I missed something that should affect my ratings.

---

> ### Author Rebuttal · Authors · 2023-08-29
>
> We would like to express our gratitude to the reviewer for providing valuable feedback. We are encouraged that Reviewer\#pxCx acknowledged our work. We have carefully considered the concerns raised and have prepared a response to address them.
>
> **Q1: Limited improvement in R@2 and R@3 on PersuasionForGood**
>
> We have the same observations as the reviewer does but the improvements are significant on DebtRiskAlert. As Figure 3 shows, the strategy-only model performs much worse than the semantic-only model on the small PersuasionForGood dataset; whereas, the strategy model alone can beat the semantic model on the large DebtRiskAlert dataset. These indicate that the strategy model cannot be sufficiently trained on such a small dataset as PersuasionForGood. We need large enough datasets to draw reliable conclusions or discover effective ideas for improving persuasive dialogues.
>
> **Q2: Non-public availability of the DebtRiskAlert dataset**
>
> Due to privacy issues, we cannot release the dialogues in the DebtRiskAlert dataset. We are trying to figure out other ways for safe sharing, e.g., releasing only token ids. Fortunately, many companies have their own logs of persuasion dialogues and thus they could reproduce or follow up on our work. In addition, building a larger public dataset with at least 20,000 dialogues is possible when our community pays more attention to the dialogues with a session-based goal.
>
> **Q3: How to solve the problem of multiple choice of strategies in an utterance**
>
> $A$ is a distribution vector indicating the probabilities of different strategies. For the candidate $u_c^+$, we apply the strategy classifier to get the distribution $A_c$, where multiple dimensions may have non-zero values. When we use our proposed Transformer decoder to predict $A_{t+1}$, which is the distribution of the next strategies, we calculate the cosine similarity between the predicted strategies and the candidate response strategies (Equation 6). If the predicted next strategies have multiple choices, the candidate with the same distribution of multiple strategies would have higher scores. We will clarify this in the revised paper.
>
> **Q4: The SOTA baseline is proposed in 2021**
>
> Different from generation-based dialogue models, retrieval-based dialogue models have not made much progress in recent years. However, retrieval-based dialogue systems are still widely used in industry, in particular when the product is required to be trustworthy. To our best knowledge, BERT-FP is still one of the best-performing methods. Some follow-up works have shown relatively small improvements over it or the improvements are not applicable to our focused problem. Our focus is on how combining the SOTA semantic models with strategy matching works for persuasive dialogue applications. Therefore, it is appropriate to choose BERT-FP as the baseline. We will add a more detailed explanation in the related work section.

---

### Meta-Review · Area_Chair_8HLd · 2023-10-04

**Recommendation:** 3

**Metareview:**

Jointly modeling of conversation semantics and strategies improves final model performance. The paper is well-written and well-motivated; however, it should incorporate suggested changes by our reviewer.

---

### Decision · Program_Chairs · 2023-10-07

**Decision:**

Accept-Findings

**Comment:**

Jointly modeling of conversation semantics and strategies improves final model performance. The paper is well-written and well-motivated; however, it should incorporate suggested changes by our reviewer.